# Using Expert Gaze for Self-Supervised and Supervised Contrastive Learning of Glaucoma from OCT Data

**Anonymous**                                                          ABC@XYZ.COM

## Abstract

In this work, we address the challenge of limited data availability common in healthcare settings by using clinician (ophthalmologist) gaze data on optical coherence tomography (OCT) report images as they diagnose glaucoma, a top cause of blindness world-wide. We use gaze data in two ways: first, we perform self-supervised pre-training via SimCLR followed by supervised fine-tuning with gaze-overlaid OCT reports; second, we directly learn gaze representations with our 'GazeFormer' model to generate pseudo-labels using a multi-task objective. We use these pseudo-labels for weakly supervised contrastive learning to detect glaucoma from a partially-labeled dataset of OCT report images. We find that self-supervised pre-training with gaze-overlaid images significantly improves glaucoma classification accuracy. Our natural language inspired region-based encoding baseline and GazeFormer model pseudo-labels enable glaucoma detection accuracy exceeding 90% even with only partially-labeled data.

**Keywords:** expert gaze data, self-supervised learning, supervised contrastive learning, glaucoma, optical coherence tomograpy

## 1. Introduction

One of the biggest challenges in artificial intelligence (AI) for healthcare lies in the acquisition of large and accurately-labeled datasets, essential for deep learning (DL). The scarcity of such labeled data hinders the development of generalizable models that can perform well on unseen data. Additionally, disparities in expert opinions, for example disagreement even among clinicians on the definition of blindness-causing eye diseases like glaucoma, can impede establishment of reliable ground truths for training. To address these issues, we use multimodal inputs: optical coherence tomography (OCT) report images and eye tracking of expert ophthalmologists as they view these OCT reports for glaucoma diagnosis. In doing so, we showcase the robust training of DL models via self-supervision derived from clinician gaze patterns.

Self-supervised learning (SSL) Chen et al. (2020) aims to learn robust representations of a data distribution and allows efficient training for downstream tasks. SSL is also robust in situations with smaller datasets with limited labels, as it enables learning of latent features that are common between different views of the same image. Medical image data is rich in patterns that may not be discernable by the human eye but that can be elucidated by the power of SSL algorithms.

Eye tracking data offers a wealth of information regarding the focus of attention and the expertise level of individuals examining medical reports. Gaze data from domain experts viewing images and videos abounds especially in medicine. Spatial gaze information encodes regions of importance, while temporal information encodes information such as order of importance. While a few methods use spatial gaze data for diagnostic AI models Li

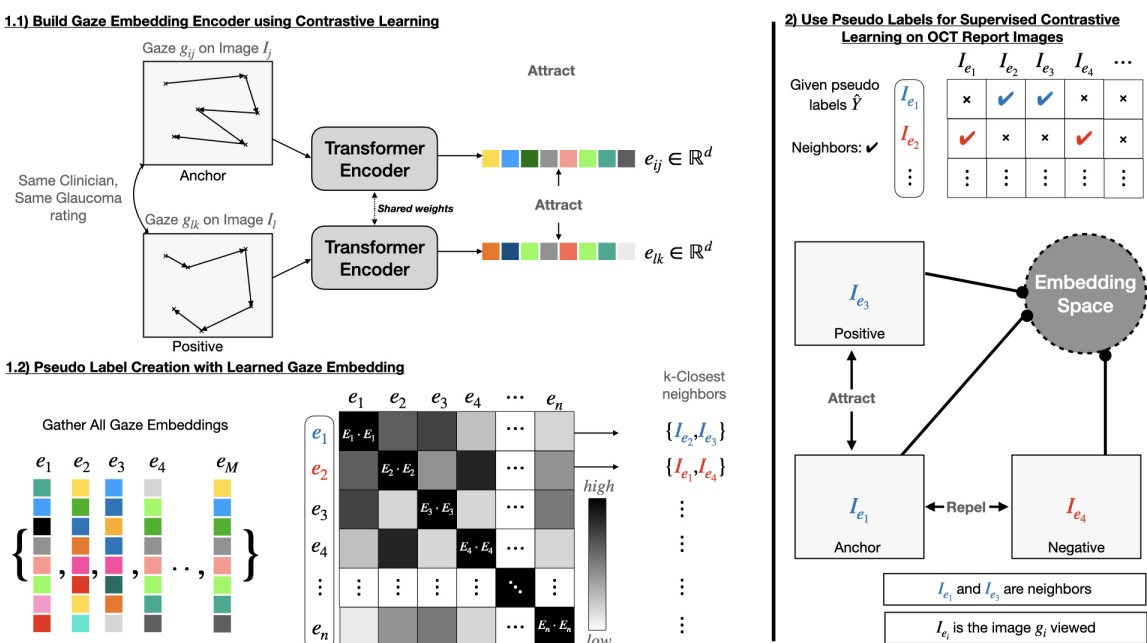

Figure 1: Overview of method: In (1.1), we take gaze data as input to pretrain GazeFormer using our multi-task objective. Then in (1.2), all embeddings are gathered to generate puesdo-labels. These labels are then used for WSupCon of images in (2). Each $e_i$ is assigned with neighbors $E$; $I_{\{e_i\}}$ now has neighbors $I_E$. More details about (1.2) are in section 3.2.3

et al. (2019) Stember et al. (2019), very few have been proposed to capture both temporal and spatial relationship explicitly to supervise downstream DL tasks. Furthermore, in contrast to supervised algorithms which require large quantities of hand-annotated or labeled data, weakly-supervised learning relies on 'inexact', coarse-grained labels (e.g., human eye-tracking) that can be more easily collected in bulk from which the label and ground truth can be inferred in place of costly expert labeling Saab et al. (2021).

In this study, we propose two methods to leverage self-supervised learning and weak-supervision, both aided by gaze data. First, we apply self-supervised pre-training on gaze-overlaid images via SimCLR Chen et al. (2020), followed by supervised fine-tuning. Second, we propose GazeFormer, a region-based eye movement encoder to learn gaze representations as pseudo-labels. These pseudo-labels are then used for Weakly-Supervised Contrastive Learning (WSupCon) based on Khosla et al. (2021) to classify OCT report images. Gaze-Former is a transformer based model Vaswani et al. (2023) and is trained with a multi-task objective that consists of triplet loss Schultz and Joachims (2003) Weinberger et al. (2005) Schroff et al. (2015) and cross entropy loss for classification. This approach leveraging eye tracking 'pseudo-labels' has the potential to enhance the performance of DL models for glaucoma diagnosis from OCT reports even with few explicit labels.

**Contributions**

- A new method to encode gaze that retains spatiotemporal relationships by modeling gaze data as words.

- **GazeFormer**, a transformer based encoder that is trained with a multi-task objective which learns robust representations. Its embeddings are used as pseudo-labels to improve model performance on glaucoma/expertise classification

- Enhanced glaucoma classification accuracy via SSL pre-training using gaze-overlaid images compared to SSL pre-training without gaze overlaid on images.

## 2. Related Work

### 2.1. Medical Expert Gaze Patterns

Expert gaze patterns have been used in different applications in machine learning and medicine. They have been found to contain useful information, such as underlying differences between expert vs. novice image viewers Brunyé et al. (2019) Akerman et al. (2023). Past work Stember et al. (2019) has also attempted to use masks generated from eye-tracking of experts while viewing radiology images vs. masks hand-annotated by experts, to compare resulting Structures of Interest (SOIs) segmented via AI. This past work showed that eye-tracking is *not significantly different* in quality to hand annotations for segmenting SOIs. This finding provided evidence that even coarse, inexact eye movements can provide the information necessary to train AI systems to achieve accurate DL-based segmentation. Other work Li et al. (2019) has attempted to specifically enhance glaucoma detection from fundus images of the retina using labeled human attention maps and region localization as well as classification convolutional neural networks (CNNs).

### 2.2. Self-Supervised Learning

In contrast to supervised learning, self-supervised learning (SSL) has shown its potential to serve as an effective pre-training strategy to learn better representations Chen et al. (2020) Caron et al. (2021), thus enabling more robust performance than supervised learning alone especially when labeled data is limited. Balestriero and colleagues Balestriero et al. (2023) offered a detailed description of state of the art methods in SSL, including BYOL Grill et al. (2020) and more. SSL has also shown success in various medical applications, such as for medical image segmentation Chaitanya et al. (2020) and for electronic health records Krishnan et al. (2022). SimCLR Chen et al. (2020) is a contrastive learning method that attempts to maximise agreement between two views of the same image through NT-Xent loss. More recently, Khosla and colleagues Khosla et al. (2021) extended NT-Xent loss introduced in SimCLR by leveraging labels during contrastive pretraining, showing labels can be incorporated into a contrastive learning framework.

### 2.3. Machine Learning on Gaze Data

Gaze data has been used in various tasks in machine learning for both weak supervision as well as gaze generation.

Gaze generation's goal is to create realistic gaze patterns that are similar to those of human viewers. Models such as recurrent neural networks (RNNs) and CNNs have been used to generate gaze patterns Li et al. (2022) Assens et al. (2017) Xia et al. (2019) Kümmerer et al. (2022) Yang et al. (2020), where gaze was modeled via a reward function and represented as 3D volumetric input.

Weak supervision seeks to use weak labels to supervise a model instead of the actual ground truth. Saab and colleagues Saab et al. (2021) extracted features from gaze data of clinicians on biomedical images, using these features as labels to aid in supervising models. This approach is most similar to our work; however, our goal is to use gaze data to aid in eye disease classification in an environment where only few labels exist.

## 3. Methods

### 3.1. Problem Setting and Approach Overview

We are given two datasets: a dataset of OCT report images with their corresponding labels $\mathcal{D}_{OCT} = \{(x_i, y_i)\}_{i=1}^N$ and a gaze dataset of clinicians' gaze on OCT reports $\mathcal{D}_{gaze} = \{(g_i, \tilde{y}_i, y_i^e, c_i)\}_{i=1}^M$. $g_i$, $\tilde{y}_i$, $y_i^e$, and $c_i$ are the gaze time series data, clinician's diagnosis (glaucoma or healthy), expertise of the given clinician, and the corresponding clinician, respectively. We are given an incomplete dataset, where a fraction of $\mathcal{D}_{OCT}$ contains ground truth labels. In total, we have 177 OCT report images and 467 eye-tracking fixations. Eye-tracking fixations were collected with Pupil Labs Core and Tobii Fusion eye-trackers while ophthalmologists viewed OCT images. Clinician experience level varied from resident to faculty; they were asked to rate each OCT report from 0 (healthy) to 100 (glaucoma). Each OCT report image $x_i$ has gazes $G_i = \{g_{i1}, g_{i2}, ...\}$ and $\sum_{i=1}^N |G_i| = M$. Each gaze $g_{ij}$ also has embedding $e_{ij}$, and $I_{\{g_{ij}\}}$ or $I_{\{e_{ij}\}}$ is the image $x_i$ that was looked at by $g_{ij}$. We will use and $g_i$ and $e_i$ only when discussing gaze data alone for brevity. We will also call the learned puesdo-labels $\hat{y}$.

Our goal is to use gaze data to learn an informative embedding that guides constrastive learning with OCT reports, thereby learning more robust representations that can improve downstream glaucoma classification performance. These learned embeddings are transformed into pseudo-labels that guide training with OCT reports. Based on observation, we hypothesize that given the same clinician, their gaze pattern should be more similar on OCT reports of the same class than on OCT reports of a different class.

Figure 1 shows the overall process of our approach. We first train GazeFormer, and use the embeddings to learn pseudo-neighbors in the form of closest neighbours; these pseudo-labels are then used for WSupCon. Finally our model is fine-tuned with labeled data for linear evaluation: classification of a report as glaucomatous or healthy.

### 3.2. Gaze Representation Learning

We take inspiration from the recent success of BERT Devlin et al. (2019) and Sentence-BERT (SBERT) Reimers and Gurevych (2019) in learning representations in natural language. Particularly, SBERT Reimers and Gurevych (2019) takes sequences/paragraphs of text and learns representations that can meaningfully compare sequences of text. Gaze data is a time-series from a participant's viewing of a given image. Gaze contains spatiotemporal

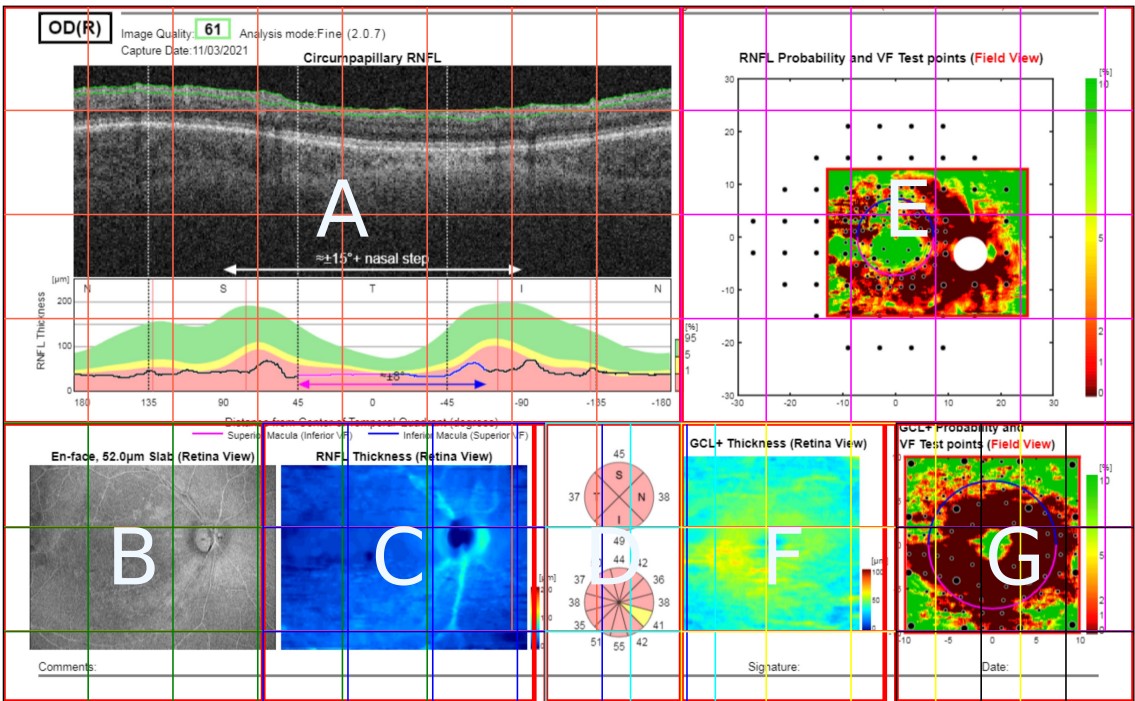

Figure 2: OCT report is split into regions, from A to G. Then each region is split into grids.

information; the order of eye fixations and duration of each fixation contain information about relative importance that leads to the participant's diagnosis decision.

Our goal is to learn a good representation $e_i \in \mathbf{R}^d$ of these sequences so we can distinguish similarities between different gazes by modelling them as words. In order to capture information from gaze data for our language-inspired DL approach, for each gaze time-series, we encode gaze in the following ways (depicted pictorially in Figure 2):

1. region-based: $g_i^{region}$, we convert each fixation to the letter that corresponds to the current region in which it falls (A, B, C, D, etc.).

2. region-based count vector: we convert $g_i^{region}$ into a count vector $g_i^{region\_cv}$, where each element is the count of fixations in that region (total length of this vector is equal to the total number of regions, which is 7 (A-G).

3. region-grid: $g_i^{rg}$, we divide each region (A-G) into sub-regions (depicted by red grids in Figure 2). Each fixation is then quantized to an integer (0-109), just as a vocabulary maps a word to an index. Since each fixation has different duration lengths, we also bin fixations into $100ms$ bins such that each element in our fixation sequence corresponds to the same amount of time. Fixations longer than $100ms$ are split into different bins and averaged.

### 3.2.1. BASELINES

We will establish baselines with $g_i^{region}$, $g_i^{region\_cv}$, and $g_i^{rg}$ via Principal Component Analysis (PCA), a Multi-layer Perceptron (MLP), and logistic regression (LogReg). SGT Ranjan et al. (2021) was used to convert variable-length gaze data into a fixed-length vector. For supervised baselines, the models were trained to classify experts vs. novices or glaucoma vs. healthy. We evaluate the performance of our baselines using K-Nearest Neighbour (kNN) clustering for glaucoma classification and visualize using T-SNE van der Maaten and Hinton (2008). PCA is used here to reduce the dimensionality of SGT's outputs.

For MLP and logistic regression using $g_i^{region}$ or $g_i^{rg}$, the learned representations are the activations before the final MLP layer and the sigmoid function, respectively.

Table 1: Different Configurations for Baseline Comparisons

|  | Method | Data Used |
|---|---|---|
| Baseline 1 | SGT $\to$ PCA | $g_i^{region}$ or $g_i^{rg}$ |
| Baseline 2 | SGT $\to$ PCA $\to$ MLP | $g_i^{region}$ or $g_i^{rg}$ |
| Baseline 3 | SGT $\to$ PCA $\to$ LogReg | $g_i^{region}$ or $g_i^{rg}$ |
| Baseline 4 | MLP | $g_i^{region\_cv}$ |
| Baseline 5 | LogReg | $g_i^{region\_cv}$ |

### 3.2.2. GAZEFORMER

We use a vanilla transformer encoder, with max sequence length $l = 768$, hidden dimension $h = 256$, 4 layers, and 4 heads. The output $f(pad(g_i^{rg})) \in \mathbb{R}^{l \times h}$ of the transformer is a collection of embeddings. However, since each input gaze has variable length, we need to create a single embedding to compare them. Similar to SBERT, we experiment with the MEAN pooling strategy, which takes the output embeddings and averages them to create MEAN embedding $e_i$. Additionally, outputs corresponding to zero-padding, CLS, and SOS tokens are masked before MEAN pooling for training and inference.

GazeFormer is trained with a multi-task objective. The first task, is contrastive triplet loss and the second task is classification for either expert or glaucoma. The combined loss to minimize is:

$$\mathcal{L}_{multi-task} = \mathcal{L}_{triplet} + \mathcal{L}_{CE} \tag{1}$$

Where $\mathcal{L}_{CE}$ is the cross entropy (CE) loss, for predicting expertise or between glaucoma vs. healthy. Glaucoma labels used here are clinician diagnoses $\tilde{y}$ for supervision and not ground truth (typically consensus of multiple clinicians). Based on the hypothesis presented in 3.1, for $g_i$, its positives are the other $g_j$ that were viewed by the same clinician given the same label. Triplet loss helps minimize the distance between an anchor and its positives, while maximizing distance to negatives.

Our objective is to minimize the following loss with $x_i^{rg}$ Schroff et al. (2015):

$$\mathcal{L}_{triplet} = \sum_i^{N_{triplets}} [\|f(x_i^a) - f(x_i^p)\|_2 - \|f(x_i^a) - f(x_i^n)\|_2 + \alpha]_+ \tag{2}$$

Where $[.]_+ = max(., 0)$ is used to ensure that when positive is closer than negative, we don't penalize the model. $x_i^a$ is the anchor, $x_i^p$ is the positive, and $x_i^n$ is the negative example. $\alpha$ is a hyperparameter used to avoid collapse, when $f(.)$ learns to map everything to $\mathbf{0}$. There are multiple ways to select triplet pairs; we select triplets using a batch-all strategy: all valid triplets are selected and averaged with only hard and semi-hard triplets. Easy triplets are those with loss less than 0; hence averaging with them would result in a very small loss. Hermans et al. (2017) provides a more in-depth discussion about triplet selections.

### 3.2.3. Obtaining the pseudo-labels:

As shown in Figure 1, after training GazeFormer, the embeddings are gathered to create pseudo-labels. First, the embeddings are used to compute the cosine similarity matrix, as shown in Figure 1 (1.2). We assume the embeddings that are similar correspond to the same class of images and correspond to the same clinician, since this is the criteria for generating positive pairs in triplet loss. Therefore, images that correspond to similar gaze embeddings should also be similar. More formally, image $x_i$ has gaze embeddings $E = \{e_j\}_{j=1}^{M(x_i)}$ [1]. Each $e_j$ has a sorted list of neighbors $E = [e_p, e_q, ...]$, which correspond to $\mathbb{I}_i = [I_{e_p}, I_{e_q}, ...]$. However, since it is not guaranteed that all images in $\mathbb{I}$ are unique (same image may be viewed by different clinicians), the top-k unique images are considered neighbors. The set of unique neighbors for each $x_i$ is $M_i$.

### 3.2.4. Weakly-Supervised Contrastive Learning

Inspired by supervised contrastive learning (SupCon), the pseudo-labels learned from gaze contrastive learning are used as weak labels in SupCon (WSupCon). After obtaining the set of neighbors $\{M_i\}_{i=1}^N$, we use OCT report images to train with the following loss, modified from $SupCon_{out}$ from Khosla et al. (2021):

$$\mathcal{L}_{Wsupcon} = -\sum_{i \in \mathrm{I}} \frac{1}{|P(i)|} \sum_{p \in P(i)} log \frac{exp((z_i \cdot z_p)/\tau)}{\sum_{a \in A(i)} exp((z_i \cdot z_a)/\tau)} \tag{3}$$

Let $i \in I = \{1, .., 2N\}$ be the index of augmented samples, and let $A(i) = I \setminus i$ be the set of indices minus the anchor. $P(i) = \{p \in A(i) : x_p \in M_i\}$ is the set of indices with the same label as the anchor within the augmented mini-batch. $\tau$ is the temperature parameter for softmax. $z_i = g(e_i)$ is the projected embedding similar to SimCLR. One important observation is that $Equation$ (3), does not enforce symmetry. For example, if $z_1$ is the current anchor and $P(i = 1) = \{z_2, z_3\}$, $Equation$ (3) does not enforce that $z_1 \in P(i = 2)$. WSupCon has the same training procedure as the SSL model mentioned in 3.3

### 3.3. Augmenting Supervised Learning with Self-Supervision and Expert Gaze Data

By leveraging Self-Supervised Learning (SSL) as a pre-training method, our model can learn intricate patterns from the data, capitalizing on various learned (pre-text) tasks, such as

---

1. $M(x_i)$ is the number of gaze data on $x_i$

predicting relationships within medical reports or reconstructing masked portions. Integrating ophthalmologist gaze data enhances this process, enabling the model to understand spatial cues from clinicians' gaze patterns. Subsequently, fine-tuning the pre-trained model with Supervised Learning (SL) on the available labeled data further refines its features for the specific clinical task.

This approach offers several advantages. First, the model is equipped with an understanding of the intrinsic data structure through SSL. Second, the inclusion of gaze data provides a nuanced perspective, enhancing the model's temporal and spatial comprehension. Third, the model's ability to generalize is significantly improved, due to the combined power of SSL, SL, and expert gaze insights.

In our study, we delved into the intricate relationship between unlabeled data and model performance when pre-training with SSL. We randomly sampled 25%, 50%, 75%, and 100% of our training data to treat as unlabeled data in our pre-training task. Then, we systematically sampled the dataset into different proportions of labeled data, ranging from 10% to 90% for our SL fine-tuning task. We used a ResNet-50 backbone and SimCLR loss Chen et al. (2020) for SSL pre-training (200 epochs), followed by SL fine-tuning (50 epochs) with cross-entropy loss for obtaining final glaucoma vs. healthy classification.

We augmented our models in two ways: one set was trained exclusively on clean OCT reports, while the other incorporated gaze fixation data (fixation information was overlaid on the image via PyGaze heatmaps Dalmaijer (2021)). This innovative augmentation strategy aimed to provide the models with an additional layer of information, particularly in situations where complete labeled data was lacking.

To ensure the reliability of our results, we implemented a robust testing procedure. The entire dataset was randomly reordered three times, and each experiment for each unlabeled data percentage was run three times and then tested. By averaging the outcomes, we accounted for any potential variability in the data and training process, ensuring the integrity of our findings. We also present the best model results from each unlabeled data percentage category (in supplementary materials).

## 4. Results

### 4.1. Gaze Representation Learning

#### 4.1.1. Baseline

Table 1 shows results from training baseline models on auxiliary tasks, expert, and glaucoma classification. Weighted kNN is used to examine the quality of the learned embeddings on glaucoma classification, since $\hat{y}$ is generated from closest neighbors. After training the baseline models to classify expertise and glaucoma, expertise accuracy was highest, so those embeddings were used to obtain kNN clusters for glaucoma vs. healthy OCT reports (last column of Table 1). Five-fold cross-validation was performed on all models to obtain model performance. After training the models on auxiliary tasks, as show in Table 1, kNN-based glaucoma classification using gaze data baseline embeddings has sub-optimal performance. Since the embeddings obtained from the expertise classification task resulted in highest performance, they are used for generating baseline pseudo-labels $\hat{y}$, which are used for WSupCon.

| Baseline Results on Expertise and Glaucoma Classification | | | | |
|---|---|---|---|---|
| | Data Used | Expertise Accuracy | Glaucoma Accuracy | KNN Accuracy (Glaucoma) |
| Baseline 1 | $g_i^{region}$ | NA | NA | 0.56 |
| Baseline 1 | $g_i^{rg}$ | NA | NA | **0.59** |
| Baseline 2 | $g_i^{region}$ | 0.64 | **0.62** | 0.51 |
| Baseline 2 | $g_i^{rg}$ | **0.75** | 0.53 | 0.57 |
| Baseline 3 | $g_i^{region}$ | 0.51 | **0.62** | 0.50 |
| Baseline 3 | $g_i^{rg}$ | 0.49 | 0.51 | 0.56 |
| Baseline 4 | $g_i^{region\_cv}$ | 0.63 | 0.56 | 0.50 |
| Baseline 5 | $g_i^{region\_cv}$ | 0.73 | 0.61 | 0.57 |

Table 1: Baseline results on auxiliary tasks and kNN with embeddings from expert classification. Note: Fields with NA could not be generated since they are the reduced embeddings from PCA.

### 4.1.2. GAZEFORMER

To evaluate GazeFormer, we employed linear evaluation and weighted kNN. Linear evaluation was performed on the pre-trained transformer model by freezing the weights and attaching a linear layer for predictions. kNN was used to evaluate the features of the model at predicting glaucoma vs. healthy clusters. Additionally, since novice's (resident's) gaze data are not as informative as that of experts (faculty) Brunyé et al. (2019) Akerman et al. (2023), we also trained with gaze data from experts only. In this case, we only evaluated with kNN. Table 2 shows the expertise and glaucoma classification accuracy using cross entropy loss applied to embeddings extracted from pathway 1.1 shown in Figure 1. Glaucoma classification accuracy is obtained using each clinician's diagnoses $\tilde{y}_i$ rather than ground truth (from consensus of multiple clinicians). We present Matthews Correlation Coefficient (MCC) to evaluate model performance inspite of dataset imbalance.

| Supervised Gaze Contrastive Learning with Cross-Entropy Objective Only | | | | |
|---|---|---|---|---|
| Loss | Data Used | Accuracy | Glaucoma KNN Acc | MCC |
| 1. Expert: $\mathcal{L}_{CE}$ | $g_i^{rg}$ | 52.12 % | 51.09 % | -0.0070 |
| 2. Glaucoma: $\mathcal{L}_{CE}$ | $g_i^{rg}$ | 76.40 % | 77.27 % | 0.5288 |
| 3. Glaucoma: $\mathcal{L}_{CE}$ | Expert $g_i^{rg}$ | **87.18 %** | **81.08 %** | **0.6429** |

Table 2: Transformer encoder trained with cross entropy only. We either train on Glaucoma classification using $\tilde{y}$ or Expert classification.

### 4.1.3. Obtaining the Pseudo-Labels

After training our transformer model with contrastive triplet loss as shown in pathway 1.1 of Figure 1, we obtained the accuracies shown in Table 3, which motivated our decision of which model to use to generate pseudo-labels for supervised contrastive learning.

| Supervised Gaze Contrastive Learning (Triplet and Multi-task Objectives) | | | | | |
|---|---|---|---|---|---|
| Loss | Data Used | Linear Eval Expert Acc | Linear Eval Glaucoma | Glaucoma KNN Acc | MCC |
| 1. $\mathcal{L}_{triplet\_loss}$ | $g_i^{rg}$ | 43.18 % | 44.71 % | 60.00 % | 0.1351 |
| 2. $\mathcal{L}_{triplet\_loss}$ | Expert $g_i^{rg}$ | NA | 43.47 % | 66.07 % | 0.3073 |
| 3. $\mathcal{L}_{multi-task}$ with Expert CE | $g_i^{rg}$ | 55.95 % | 50.00 % | 55.13 % | 0.0786 |
| 4. $\mathcal{L}_{multi-task}$ with Glaucoma CE | $g_i^{rg}$ | **64.44** % | 72.82% | 79.35 % | 0.5847 |
| 5. $\mathcal{L}_{multi-task}$ with Glaucoma CE | Expert $g_i^{rg}$ | NA | **88.46** % | **88.24** % | **0.7571** |

Table 3: Evaluating pre-trained transformer using contrastive learning on linear evaluation and kNN. For each loss, we used a linear layer on top of the transformer to train the linear layer, and we used both expertise and expert diagnoses $\tilde{y}$ as target. For models 2. and 5., since only expert data was used, both train and test would only consist of expert data.

Table 3 shows that the best pseudo-label performance is achieved by the model trained on multi-task loss with glaucoma classification cross-entropy using expert-only gaze data. Thus, pseudo-labels generated from this model are used for WSupCon.

### 4.1.4. Weakly-Supervised Contrastive Learning

Now we will present our results of using pseudo-labels generated using gaze from the previous section. We validated our framework using our dataset of OCT report images. We will compare our WSupCon approach to our baseline, which is pseudo-labels generated using Baseline 2 (see section 4.1.1).

Figure 4 shows linear evaluation results with pseudo-labels $\hat{y}$ from baseline vs. those created via gaze contrastive learning. For both baseline and gaze contrastive learning, we use different amounts of data out of the trainset to pre-train with WSupCon. Then, for each model, we also varied the amount of labeled data used to train (fine-tune) the fully connected layer. We trained using WSupCon for 200 epochs and fine-tuned the linear layer with 50 epochs for all models. The overall procedure is shown in Figure 3.

### 4.2. Augmenting Supervised Learning with Self-Supervision and Expert Gaze Data

In our analysis, Kolmogorov-Smirnov tests confirmed non-normality in all models' accuracy distributions (p=0), emphasizing the unique nature of the data. Subsequent Mann-Whitney

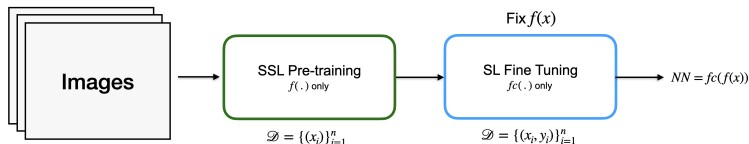

Figure 3: Self-Supervised Learning Training Pipeline. $f(.)$ is the encoder and $fc(.)$ is the fully connected layer. Pre-training stage only takes images; then the encoder's weights are frozen for fine-tuning.

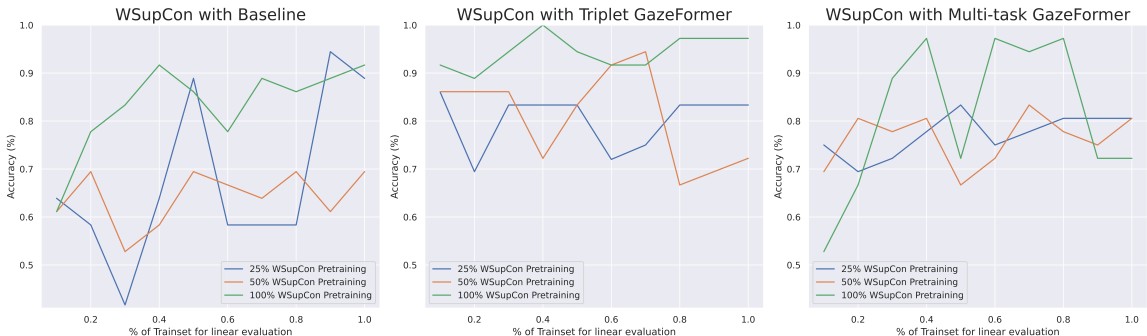

Figure 4: Glaucoma detection accuracy on WSupCon with varying % pre-training data using pseudo-labels generated from baseline and GazeFormer, triplet model 2. and multitask model 5. from Table 3 (plots show best of 3 random re-orderings of test data; averages are shown in supplementary materials).

U tests of the average accuracy results between with and without gaze fixations yielded p-values of 0.0003873, 0.0003792, and 0.01706 for 25%, 50%, and 100% SSL pre-training, respectively (75% is excluded since its p-value was insignificant). This indicates a significant difference in accuracy between models with and without gaze fixation data. These results emphasize the impact of integrating gaze fixation data on glaucoma classification accuracy from OCT reports, highlighting its potential to enhance model performance compared to a model trained solely on clean OCT reports.

## 5. Discussion

### 5.1. Baselines

Our baseline results indicate that although expertise classification reached 75% accuracy, glaucoma classification and kNN accuracy barely exceeded chance when embeddings were derived from SGT, PCA, MLP, or logistic regression approaches. However, the fact that these baselines crossed the chance threshold validates our encoding scheme using language-inspired region-based, region-based count vector, and region grid methods.

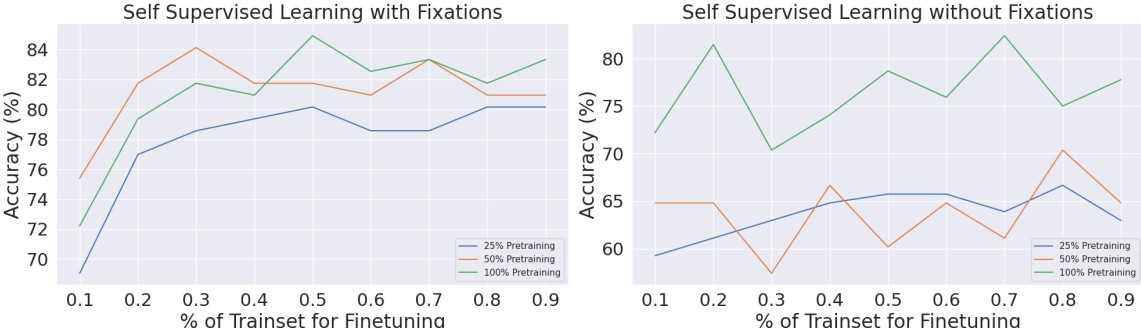

Figure 5: Comparison of average glaucoma detection accuracy of models trained using varying amounts of OCT report data for SSL pre-training, followed by supervised fine-tuning with 10% to 90% labeled data, with gaze data superimposed (left) or without gaze data (right).

## 5.2. Gaze Contrastive Learning

### 5.2.1. GAZEFORMER

Gaze contrastive learning has shown to be effective as it achieves the best results compared to baseline and when only trained with cross entropy. The multi-task nature of our objective further encourages the model to bring gaze embeddings closer together. Although the multi-task objective with glaucoma classification cross entropy and expert data only (model 5., Table 3) outperforms that trained with a cross-entropy objective alone and only expert data (model 3., Table 2), the performance gap is not significant (88.46% vs. 87.18%). This may be due to the triplet loss being noisy, since triplets are selected at each training step from the current mini-batch. The triplet selection strategy could also affect performance, as Hermans et al. (2017) found in their work that the batch-hard strategy works best. Experimenting with a weighted sum between triplet loss and cross entropy loss could also help optimize the benefits from both losses. Lastly, since past work has shown Akerman et al. (2023) that gaze data can be used to classify expertise with accuracy beyond 90%, this suggests our current training method may be improved by learning more generative patterns in gaze data. We could extend our approach by training GazeFormer with additional tasks, such as masked language modeling and next-word prediction, effectively learning a generative model for gaze data. This could be done with a window of masked input in order to predict the next few fixations.

### 5.2.2. OBTAINING THE PSEUDO-LABELS

We generated pseudo-labels from gaze contrastive learning and multi-task loss (including triplet loss combined with glaucoma cross-entropy loss) and applied those for weakly supervised supervised contrastive learning from OCT report images. These pseudo-labels achieved up to 88.46% linear glaucoma classification accuracy. Further experiments could use k-means clustering or other techniques to assign labels.

### 5.2.3. Weakly-Supervised Contrastive Learning

WSupCon with GazeFormer generated pseudo-labels outperforms baseline for each variation of the model, where a different % of the trainset data was used for WSupCon pre-training as shown in Figure 4. There are occasional drops in testset accuracies, and these could be explained by looking at the t-SNE plots of both gaze embeddings and image embeddings (shown in supplementary materials). Since we are only training a single linear layer, small perturbations might result in incorrect classifications. Triplet (model 2., Table 3)) and multi-task (model 5., Table 3) objectives perform comparably across varying fractions of the pre-training (25%, 50%, 100%), with the triplet loss accuracy exceeding 95% at 100% WSupCon pre-training and 40% labeled data.

### 5.3. Augmenting Supervised Learning with Self-Supervision and Expert Gaze Data

Our study yielded significant insights as seen in Figure 5. Notably, at moderate to high levels of unlabeled data usage for SSL pre-training, models trained both with and without gaze fixation data exhibited superior performance to the models trained with a smaller amount of unlabeled data. In addition, as seen in Figure 5, the comparison of average results between models trained on the % unlabeled data with (left) vs. without (right) fixations demonstrate an ability to achieve impressive results with substantially less labeled data for fine-tuning. In fact, even a model using 25% of the training data with fixations for SSL pre-training was able to achieve better or comparable results to a model using 100% of the training data without fixations. This finding suggests that the inclusion of gaze fixation information served to boost the model's confidence. Fixation data enabled the model to effectively leverage the available unlabeled data during this Self-Supervised Learning (SSL) task, offering a potential solution to the common challenge in healthcare settings of lack of access to completely labeled data.

## 6. Conclusions and Future Directions

Our work showcases that medical expert gaze data (specifically eye movements of ophthalmologists as they view optical coherence tomography reports for glaucoma detection) has the potential to enhance disease detection accuracy especially in settings when access to labeled data is lacking. This was made evident by the significant improvement in accuracy when we utilized a self-supervised pre-training strategy to detect glaucoma from OCT reports superimposed with fixation heatmaps followed by supervised fine-tuning with partially-labeled data. Pseudo-labels generated purely from region-based encodings of gaze data on OCT reports enabled downstream glaucoma classification with suboptimal accuracy; however, pseudo-labels derived from supervised gaze contrastive learning (GazeFormer) using cross-entropy, triplet, and multi-task loss achieved up to 88.46% accuracy. Weakly supervised contrastive learning using these pseudo-labels was effective at achieving accuracy up to and beyond 95% when labeled data and pre-training dataset size were optimized. Future directions of this work include employing alternate embedding approaches by quantizing eye tracking into features based on characteristics beyond region alone, pre-training on large datasets of eye tracking data (even those of non-experts) prior to fine tuning on clinician

data, and leveraging text responses of clinicians in addition to their eye movements, enabling the use of large language model inspired methods to predict the next fixation in a sequence, much like predicting the next word in a sentence.

## Acknowledgments

Acknowledgements left out for now for anonymity of review.

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
