# OpenReview forum: "Using Expert Gaze for Self-Supervised and Supervised Contrastive Learning of Glaucoma from OCT Data"
_NeurIPS.cc/2023/Workshop/Gaze_Meets_ML — Submitted to Gaze Meets ML 2023_

### Official Review · Reviewer_ZifK · 2023-10-12
**Using Expert Gaze for Self-Supervised and Supervised Contrastive Learning of Glaucoma from OCT Data**

**Rating:** 6
**Confidence:** 5

**Review:**

The authors claim to use clinicians' gaze data in diagnosing glaucoma from OCT images and propose it as an effective solution to mitigate issues arising from limited data availability in medical computer vision. However, several queries need to be answered to convey clarity to the readers.
•	Have other methods incorporating gaze data been explored previously? How does this method differentiate from or improve upon them?
•	The challenge of limited data availability in healthcare settings is mentioned. Could the authors provide a more detailed overview of how much data is typically available and how their method alleviates this issue? What are the current challenges in glaucoma detection using OCT and how does the incorporation of expert gaze data address these challenges?
•	Were there any external datasets or a separate validation set used to test the generalizability of the proposed method?
•	Are the improvements in glaucoma classification accuracy statistically significant, especially when compared to baseline models or other methods? How were statistical tests chosen, and were the results corrected for multiple comparisons if appropriate?
•	How robust is the proposed method to variations in gaze data quality or inconsistencies among different ophthalmologists? Was the method tested in real-world clinical settings, and if so, how did it perform?
•	Can the authors provide more technical details about the ‘GazeFormer’ model, its architecture, and its training process? How do you handle varying lengths of gaze sequences when encoding them as words? Is there a fixed length or some form of padding involved?
•	Can you provide more detail on how gaze data's spatiotemporal relationships are retained when modeled as words? What are the advantages of this approach over other potential encoding schemes?
•	How does the multi-task objective of GazeFormer aid in learning more robust representations of the gaze data compared to single-task objectives?
•	Could the authors provide insights into the architecture details of the GazeFormer, such as the number of layers, attention heads, and any specific adaptations made to the standard transformer architecture?
•	How do the embeddings from GazeFormer, used as pseudo-labels, compare in quality to human-annotated labels, and how do they help in improving model performance?
•	Were there any instances where the pseudo-labels generated by GazeFormer led to misclassification or sub-optimal performance? If so, how did the authors address these issues?
•	How does the glaucoma classification accuracy vary across different amounts of gaze data? For instance, is there a saturation point where adding more gaze data does not lead to significant improvements in accuracy?
•	Do the authors believe the gaze data encoding and GazeFormer method could apply to other medical imaging tasks or even non-medical tasks?
•	What are the potential future extensions of this work, especially concerning the encoding of gaze data and its application to other domains?

---

### Official Review · Reviewer_Yx21 · 2023-10-19
**Noteworthy application of gaze data and self-supervised learning for glaucoma detection**

**Rating:** 8
**Confidence:** 4

**Review:**

This work proposes a self-supervised pretraining method to improve glaucoma detection accuracy from OCT reports. The Gazeformer uses a transformer model and a sequence of gaze data regions on the OCT image to output a single mean-pooled embedding vector, trained using multi-task objectives through contrastive learning and auxiliary classification tasks. The Gazeformer is further used to generate pseudo labels for weakly supervised contrastive learning from original or heatmap overlaid OCT report images, followed by supervised fine-tuning for glaucoma detection.

This paper is interesting and important in the medical area, considering the challenges of labeled data.

The paper is well organized and provides sufficient experimental details and discussions. Authors evaluate each stage of their pipeline and also provide necessary ablations.

Minor comments:
1. In section 5.2.2, the third line, “supervised” is repeated twice
2. Remove “:” from section 3.2.3 heading

---

### Official Review · Reviewer_UXXx · 2023-10-21
**Review of sing Expert Gaze for Self-Supervised and Supervised Contrastive Learning of Glaucoma from OCT Data**

**Rating:** 4
**Confidence:** 2

**Review:**

The authors have reported using eye gaze tracking in OCT image classification with limited data. They report two contributions: 1) An interesting and potentially novel method of representation learning inspired by BERT to create a representation of eye gaze data (location/duration of fixation time series) to be used in ML models for other tasks. 2) GazeFormer a transformer based encoder to learn representations from gaze data.

Concerns: The paper is extremely long and complicated. For reasons that I do not understand, the authors have decided to combine two papers in one. Throughout, it is is extremely difficult to keep track of the relevance of concepts to the different methods. The authors have chosen a ‘linear’ approach to writing a sequence of sections related to the two methods.

While I really like the first contribution, after an hour of struggle, I was unable to understand the relationship between the contributions and my current (potentially wrong) understanding is that there is none.

Accepting the paper in its current form is a dis-service to authors. I suggest the authors break this into two well written and well argued papers.

---

### Meta-Review · Area_Chair_cBoM · 2023-10-26

**Recommendation:** Reject
**Confidence:** 4

**Metareview:**

Reviewers had concerns about the paper's clarity and organization and the effectiveness of the gaze data in detecting glaucoma in OCT. They asked for more information on the limited data availability challenge, the generalizability of the proposed method, the accuracy improvements, and the robustness of the method to variations in gaze data quality. They also requested more technical details on the GazeFormer model. Lastly, they inquired about the variation in glaucoma classification accuracy with different amounts of gaze data and the applicability of this method to other imaging tasks.

I recommend reorganizing the paper per the reviewer's suggestions and addressing their concerns.

---

### Decision · Program_Chairs · 2023-10-26

Reject